# In Vitro Study on Bone Heating during Drilling of the Implant Site: Material, Design and Wear of the Surgical Drill

**DOI:** 10.3390/ma13081921

**Published:** 2020-04-19

**Authors:** Juan Carlos Bernabeu-Mira, Hilario Pellicer-Chover, Miguel Peñarrocha-Diago, David Peñarrocha-Oltra

**Affiliations:** Oral Surgery Unit, Department of Stomatology, Faculty of Medicine and Dentistry, University of Valencia, Oliag Gascó 1, 46010 Valencia, Spain; hilariopellicer@gmail.com (H.P.-C.); miguel.penarrocha@uv.es (M.P.-D.); david.penarrocha@uv.es (D.P.-O.)

**Keywords:** bone heating, drilling, implant site, design, material, wear

## Abstract

Objective: An in vitro study was made to compare mean thermal variation according to the material, design and wear of the surgical drills used during dental implant site preparation. Material and methods: Three study groups (stainless steel drills with straight blades; diamond-like carbon-coated drills with straight blades; and diamond-like carbon-coated drills with twisted blades) were tested to compare material, design and wear of the surgical drill in terms of overall mean values (complete sequence of drills) and specific mean values (each drill separately). The groups comprised four drills: initial, pilot, progressive and final drill. Implant site configuration was performed through an intermittent and gradual drilling technique without irrigation at 800 rpm in standardized synthetic blocks. Maximum axial loading of two kilograms was controlled by an automatic press. Each surgical drill was submitted to 50 drillings and was sterilized every five uses. A thermographic camera analyzed the mean thermal changes. The software-controlled automatic press kept systematic drilling, axial loading and operational speed constant without any human intervention. Student’s t-test, ANOVA and multiple linear regression models were performed. The level of significance was 5% (p = 0.05). Results: The overall mean comparison between the stainless steel and diamond-like carbon-coated materials showed no statistically significant differences (p > 0.05), though specific mean comparison showed statistically significant differences between the drills of the different groups (p < 0.05). The twisted blades exhibited less overall and specific mean thermal variation than straight blades for the progressive and final drills (p < 0.01). In addition, the initial and pilot drills showed a greater mean thermal change than the progressive and final drills. The mean thermal variation was seen to increase during the 50 drillings. Conclusions: Within the limitations of this study, it can be concluded that the drill material did not significantly influence the overall mean thermal variation except for the pilot drill. The drill design affected overall and specific mean thermal variation since the twisted blades heated less than the straight blades. The initial and pilot drills increased the specific mean thermal variation with respect to the progressive and final drills. In addition, all drills in each group produced a gradual increase in mean temperature during the 50 drillings.

## 1. Introduction

Standard protocols in implant dentistry involve the use of surgical drills to prepare implant sites. This generates an increase in temperature due to friction between the surgical drill and bone.

According to the literature, thermal osteonecrosis of the peri-implant bone may occur if the temperature stays above 47 °C for one minute [1]. Controlling thermal trauma to bone due to drilling of the implant site is necessary to ensure survival of all cells and proteins implicated in the bone remodeling [2,3]. Bone overheating during drilling may cause early implant failure due to the accumulation of necrotic bone, fibrous tissue, bone sequestration, bacteria and inflammatory infiltrates around the implant [4]. To avoid this, most implant systems use irrigation as part of the drilling protocol. However, a randomized clinical trial has suggested that drilling without irrigation could be viable and comparable to drilling with irrigation in terms of bone loss and early implant failure at one year of follow-up [5]. This drilling technique without irrigation allows better visualization of the drilling area and better control of the drill and of bone harvesting. Therefore, optimizing the drilling system as much as possible to control thermal increments without the need for irrigation systems makes sense in order to keep bone below the critical temperature and thus prevent bone necrosis.

Many factors are involved in the thermal changes during drilling of the implant site: rotational speed, load or pressure, drilling technique, drilling depth, irrigation, drill material, drill design, drill wear and characteristics of the bone [6,7,8].

Thermal changes have been studied for a great variety of drill materials. The most widely studied material is stainless steel [6]. However, ceramics [9] and coated stainless steel such as diamond-like carbon-coated [10] or direct diamond-coated drills [11] have also been studied. The thermal changes associated to diamond-like carbon-coated drills have been investigated in a study [10] that found no statistically significant differences with respect to stainless steel drills.

The possible thermal changes have been analyzed in relation to certain design characteristics of the drills, such as the tip angle [12], cutter shape [13], number of cutting edges [12], magnitude of bone contact surface [14] and length of the active part [15]. However, none of the studies published to date have compared the temperature changes between straight and twisted blades.

Wear due to repeated use of the surgical drill has been recorded thermally in a number of studies [12,16], because the cutting efficiency and quality of conventional rotating drills for bone implant surgery would be affected. Thus, hard coatings are generally applied on the blades of these surgical drills, especially when precise cuttings and insert stability are required. An example is the diamond-like carbon-coated drill. This coating would improve cutting efficiency due to its greater hardness compared to stainless steel [10,11].

The present study was carried out to compare the mean thermal differences between stainless steel and diamond-like carbon-coated drills, and between straight and twisted drill blades—and to analyze the thermal evolution of the drill during repeated use. The working hypothesis was that the material, design and wear of the surgical drill can influence mean thermal variation during drilling of the implant site.

## 2. Material and Methods

### 2.1. Study Design

The present in vitro study on the mean thermal changes during drilling of the implant site was carried out at the Research, Development and Innovation (I+D+I) Department of Ticare (Mozo-Grau, SA, Valladolid, España). No ethical approval was required since the study was performed on a standardized synthetic block. The study was reported following the CRIS (checklist for reporting in-vitro studies) [17].

Three implant drilling systems (Ticare, Mozo-Grau, SA, Valladolid, España) were evaluated (Table 1): stainless-steel drills with straight blades (SS-SB); diamond-like carbon-coated drills with straight blades (DLCC-SB); and diamond-like carbon-coated drills with twisted blades (DLCC-TB). The surgical drilling sequence comprised four drills in each group (Table 1).

The drills of the SS-SB and DLCC-SB groups were produced with an identical design and only differed in the material component of the drills. In turn, the drills of the DLCC-TB and DLCC-SB groups were made of the same material and only differed in terms of the design of the blades for the progressive and final drills. Statistical comparison between the SS-SB and DLCC-SB groups yielded the results with respect to drill material, and the comparative analysis of the DLCC-SB and DLCC-TB groups evaluated the results with respect to drill design. It is important to note that the initial and pilot drills of the DLCC-SB and DLCC-TB groups were identical in all aspects.

### 2.2. Operational Procedure

The experimental parameters are shown in Table 2. All the operational variables were kept constant by the automatic press (ZwickRoell Z 2.5^®^, ZwickRoell, Ulm, Germany) equipped with a software system (ZwickRoell testXpert II V3.2^®^ ZwickRoell, Ulm, Germany). This machine applied constant axial loading to the drill through the head of contra-angle, controlling maximal axial loading, the drilling technique, drilling depth and operational speed (mm/min). This automated system was designed to perform multiple drillings in a reproducible, precise and controlled manner, without the direct intervention of an operator (Figure 1).

Each drill was used to perform 50 drillings with a total sample (n) of 600 drillings. The overall mean thermal variations between the SS-SB and DLCC-SB groups was measured as the mean of 200 drillings. The mean thermal variations between the DLCC-SB and DLCC-TB groups was measured as the mean of 100 drillings, since the initial and pilot drills were of the same design and material.

The drills were held by the non-active part through the corresponding support in order to preserve its integrity. In addition, the drills were sterilized every 5 uses. This sterilization process lasted a total of 30 min (5 min sterilization, 10 min drying and a vacuum phase) at a temperature of 134 °C, 2 bars of pressure and 2.5 kg of maximum loading.

### 2.3. Standardized Synthetic Blocks

The drillings were carried out on standardized synthetic block (Implant training plate, Biomechanicals Models^®^, SelModels SL, Barcelona, Spain) with dimensions of 140 × 90 × 15 mm. The block was formed by a constant cortical plate of 2 mm of thickness which covered the cancellous part to simulate human bone conditions type 2 according to the Lekholm and Zarb classification [18]. This structure guaranteed the equivalence of the different drilling sites respect to the density and the ratio between the cortical and cancellous portions [19]. Thermal conductivity of the standardized synthetic block (0.3 W/m/K) was analogous to the human bone (cortical bone: 0.29 W/m/K and cancellous bone: 0.16–0.34 W/m/K [20]). These thermal conductivity values assured that the measured thermal variation in this block was equivalent to those in human bone [21]. The room temperature was kept constant at 29 °C.

### 2.4. Thermal Measurements

The FLIR E60bx ^®^ thermographic camera (FLIR Systems OÜ, Harju, Estonia) measured the mean thermal changes. Its main features were image quality or thermal resolution 320 × 240 pixels, recordable temperature range −20 °C to +120 °C and thermal sensitivity < 0.045° pixels or < 50 mK NETD (noise equivalent temperature difference). This method has been used in other in vitro studies related to this topic [22,23,24].

To the effects of data collection, we calculated the thermal variation (ΔT) between the maximum temperature (T_m_) and the initial temperature (T_0_), according to the mathematical expression: ΔT = T_m_ − T_0_. Thermal variation was expressed in degrees centigrade (°C). The data were collected from video formats in order to be able to evaluate the thermal changes through a continuous temporal analysis. Such data collection was conducted by an independent researcher.

The overall mean thermal variation was the sum of all the thermal variations of all the drills of each respective group and was calculated to compare the design and material. The specific mean thermal variation was the sum of all the thermal variations of each drill within its respective group and was utilized to contrast comparisons between all the drills. 

### 2.5. Statistical Analysis

The Student t-test for independent sample was used to compare overall mean thermal variation according to the material and to the design. One-way analysis of variance (ANOVA) was used to evaluate the thermal effect between the drills of each group (specific mean thermal variation). Multiple comparisons were made with the Bonferroni test to avoid propagation of type I error. The thermal variation of drill wear over the number of uses was studied using multiple linear regression models. The level of significance used was 5% (p = 0.05). For a Student t-test with a confidence level of 95% and considering an effect size of 0.35 (small-medium), the resulting statistical power was 0.94.

## 3. Results

The results of the statistical analysis are presented in Table 3 and Figure 2.

### 3.1. Drill Material

There were no statistically significant differences in the overall mean values (p > 0.05) between the SS-SB and DLCC-SB groups. The specific mean thermal variations were significantly greater for the pilot drill in the SS-SB group than in the DLCC-SB group and no statistically significant differences were observed among the other drills (Table 3 and Figure 2).

### 3.2. Drill Design

Statistically significant differences (p < 0.01) were observed between the straight and twisted drill designs regarding the overall and specific mean thermal variations of the progressive and final drills (Table 3 and Figure 2).

### 3.3. Drill Type

The specific mean thermal variation of the initial and pilot drills was significantly greater than that of the progressive and final drills (p < 0.05) in all groups (Table 3 and Figure 2).

### 3.4. Drill Wear

Thermal variation increased according to a linear function over the 50 uses, independently of the drill material (Figure 3) and design (Figure 4).

## 4. Discussion

The present in vitro study analyzed different variables that affect thermal variation during drilling of the implant site, such as the material, design and wear of the surgical drill. This subject is of crucial importance since overheating of the implant site may lead to thermal osteonecrosis [1] and early dental implant failure [4].

In our study, the overall mean thermal difference between the stainless steel and diamond-like carbon-coated drills was not statistically significant. Other authors have compared others types of materials and have reported no statistically significant changes in mean thermal variation during drilling [25,26,27]. Hochscheidt et al. [10] reported a higher mean temperature with diamond-like carbon-coated drills than with stainless steel drills—though the difference failed to reach statistical significance. The mentioned study drilled bovine ribs with a different cortical thickness of between 2–3 mm, and the temperature was monitored with a thermocouple, which may affect the measurements [21,22,23,24,25,26,27,28]. The present study overcame the limitations of the thermal measurement method with infrared thermography and the of drilling medium with the use of standardized synthetic blocks [21,22,23,24,25,26,27,28]. The thermal conductive discrepancy between the tested materials could cause different heat concentrations in the friction zone [22] and distinct dissipation of the heat through the drill [10,24,29]. However, these factors do not appear to be sufficiently influential to result in statistically significant differences in overall mean thermal variation between the tested materials.

The cutting design of the drill blades showed statistically significant differences in our study, with twisted blades producing less mean thermal variation than straight blades. These significant differences can probably be explained by higher cutting efficiency and better removal of the bone chips with twisted drills [14,15]. No previous studies have compared these blade designs; direct comparisons are therefore not possible.

In the present study, drill wear significantly increased the mean thermal changes after 50 drillings and 10 sterilization cycles in all groups. The thermal change increments in this study were significantly different, in concordance with the literature [30,31]. A study [31] has analyzed wear based on scanning electron microscope analysis, revealing that repeated drill use deteriorates the physical and mechanical surface properties of the surgical drill. This mean thermal increase occurred in all types of materials [9,10,26,32]. The 10 sterilization cycles could be influenced by thermal variations, as evidenced by another study with the use of electron microscopy [33].

In our study, the mean thermal change of the initial and pilot drills increased significantly compared to the progressive and final drills, as also reported by other studies [34,35]. The first drills (initial and pilot) of the stepwise drilling procedure drill most of the material laterally and in depth, while the last drills of the sequence (progressive and final) only drill a smaller amount of material laterally [34,35]. However, other authors have reported that the last drills of greater diameters conferred greater thermal changes [24]. The pilot drill of the stainless-steel system showed a greater increase in mean thermal variation than the pilot drill of the diamond-like carbon-coated system. Differences between the tested materials in terms of hardness [10], cutting efficiency [16] or thermal conductivity [25,29] seem to be more important in the most demanding heat-related drilling phases such as the pilot drill (with a combination of lateral drilling and up to maximum drilling depth). These differences may not have a significant impact in less demanding situations such as when the initial, progressive and final drills are used. More studies are needed to elucidate the cause of the lesser thermal effect of the pilot drill, which is the greatest generator of heat.

Three methodological aspects are important to ensure the reproducibility and accuracy of tests of this kind: the drilling medium, the thermal measurement method and automatization of the operational procedure. Regarding the drilling medium, most previously published studies used several drilling media, such as bovine ribs with dissimilar densities and relationships between the cortical and cancellous bone. This fact could affect the resulting thermal variations [21]. Some studies have tried to eliminate this limitation through the use of synthetic blocks [14,23]. This drilling media has a uniform density, equivalent mechanical characteristic [14] and thermal conductivity to the human bone density [21]. Regarding thermal measurement, the most commonly used methods have been thermocouples and thermography [6]. Harder et al. [28] compared thermocouples and thermography in an in vitro study involving a total sample size of 160 drillings. They concluded that thermography more accurately reflected the intraosseous thermal changes during implant site preparation versus thermocouples, including when irrigation methods are used. However, interpositioning between irrigation and the thermographic camera could interfere with thermal measurement in the drilling zone [36]. A limitation that distances our study from the real life standard clinical drilling protocols is the absence of an irrigation method due to the use of infrared thermography [37]. Regarding automatization of the operational procedure in this study, some authors have adopted the same procedure using automatic machines without direct human intervention in order to ensure constant operational speed (mm/min), axial loading and timing during all drillings [21,28].

## 5. Conclusions

Within the limitations of this study, it can be concluded that the drill material did not significantly influence the overall mean thermal variation, except for the pilot drill. The drill design affected overall and specific mean thermal variation since the twisted blades heated less than the straight blades. The initial and pilot drills increased the specific mean thermal variation with respect to the progressive and final drills. In addition, all drills in each group produced a gradual increase in mean temperature during the 50 drillings.

## Figures and Tables

**Figure 1 materials-13-01921-f001:**
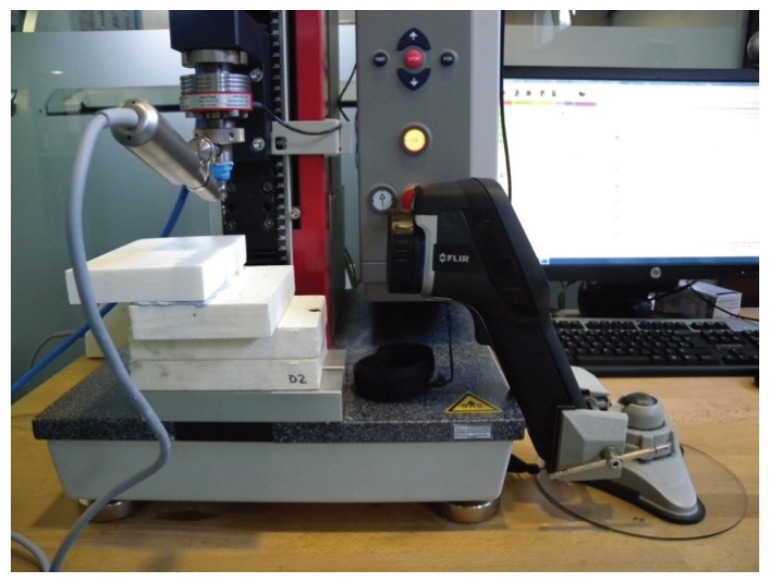
Experimental setup: automatic press (ZwickRoell Z 2.5^®^, ZwickRoell, Ulm, Germany), standardized synthetic polyurethane blocks, FLIR E60bx^®^ thermographic camera (FLIR Systems OÜ, Harju, Estonia) and drilling instruments.

**Figure 2 materials-13-01921-f002:**
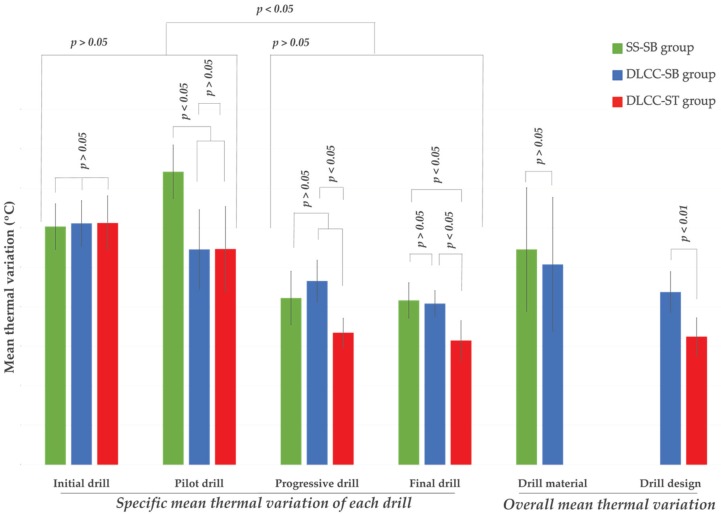
Graphic representation of the statistical analysis. The mean thermal variation of each drill and the overall mean thermal variation for each drill material and design are shown.

**Figure 3 materials-13-01921-f003:**
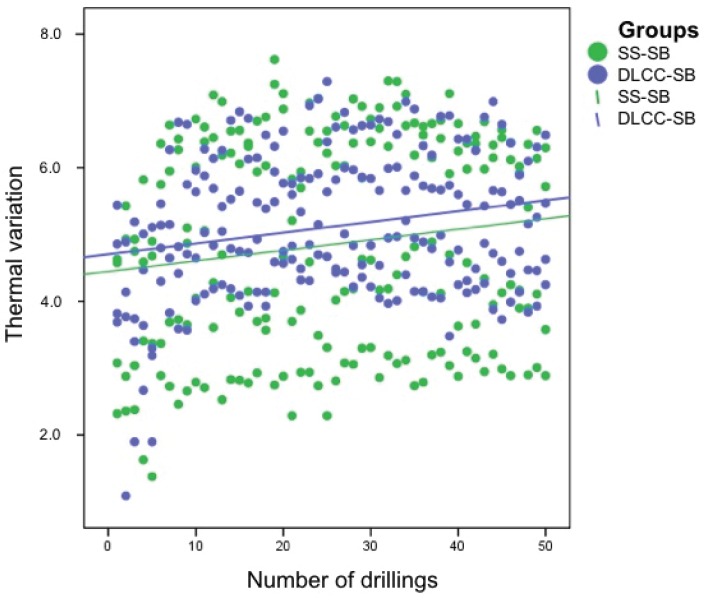
Thermal evolution over 50 drillings for the drills of the SS-SB and DLCC-SB groups. The green dots represent the thermal variation of each drill of the SS-SB group, and the blue dots represent the thermal variation of each drill of the DLCC-SB group. The green and blue lines correspond to the SS-SB group and DLCC-SB group, respectively, and represent multiple linear regression models of the thermal evolution of these groups.

**Figure 4 materials-13-01921-f004:**
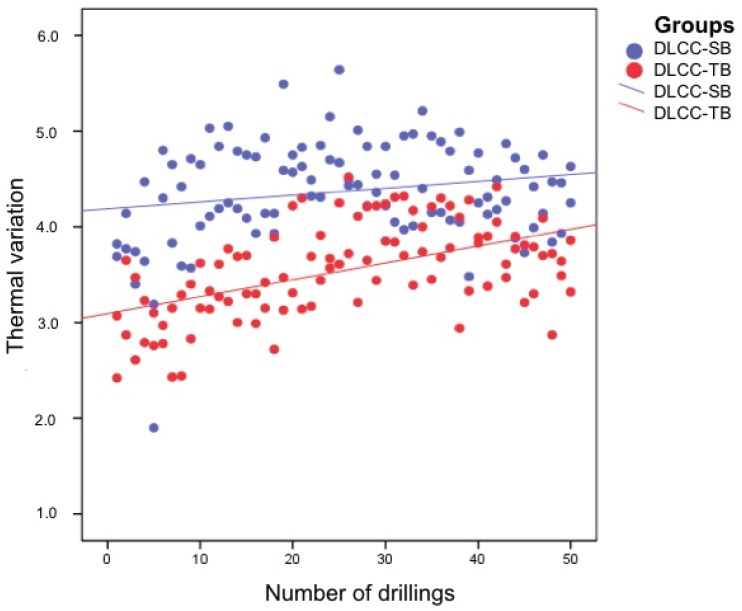
Thermal evolution over 50 drillings for the drills of the DLCC-SB and DLCC-TB groups. The blue dots represent the thermal variation of the progressive and final drills of the DLCC-SB group, and the red dots represent the thermal variation of the progressive and final drills of the DLCC-TB group. The blue and red lines correspond to the DLCC-SB group and DLCC-TB group, respectively, and represent multiple linear regression models of the thermal evolution of the progressive and final drills of these groups.

**Table 1 materials-13-01921-t001:** Study groups and design characteristics of the surgical drills.

SS-SB Group	DLCC-SB Group	DLCC-TB Group
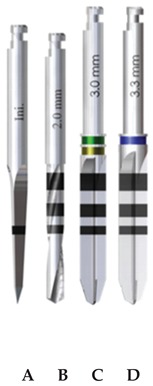	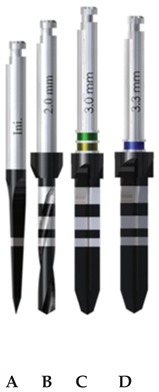	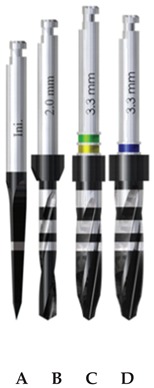

(A) Initial drill: 1.5–2 mm ∅, tip angle 30° and 3 cutting edges; (B) pilot drill: 2 mm ∅, tip angle 120° and 2 cutting edges; (C) progressive drill: 3 mm ∅, 120° tip angle and 4 cutting edges; (D) final drill: 3.3 mm ∅, 120° tip angle and 4 cutting edges(∅: diameter).

**Table 2 materials-13-01921-t002:** Description of the experimental parameters.

Experimental Parameters	
Parameter	Settings
Irrigation	None
Operational speed	50 mm/min
Rotation speed	800 rpm
Maximum axial load	2 kg
Maximum torque	45 Ncm
Depth of drilling	11.5 mm
Drilling Technique	Intermittent and gradual
Pumping	4- and 8-mm depth
Sterilization	Every 5 drillings

**Table 3 materials-13-01921-t003:** Mean thermal variations of the stainless-steel drills with straight blades (SS-SB), diamond-like carbon-coated drills with straight blades (DLCC-SB) and diamond-like carbon-coated drills with twisted blades (DLCC-TB) groups.

	Groups	SS-SB	DLCC-SB	DLCC-TB
Δ T Mean (°C)				
Specific mean Δ T of initial drills		6.03 ± 0.58	6.11 ± 0.58	6.12 ± 0.69
Specific mean Δ T of pilot drills		7.42 ± 0.68	5.45 ± 1.01	5.46 ± 1.08
Specific mean Δ T of progressive drills		4.22 ± 0.68	4.65 ± 0.53	3.34 ± 0.37
Specific mean Δ T of final drills		4.16 ± 0.45	4.08 ± 0.33	3.14 ± 0.51
Overall mean Δ T for drill material		5.45 ± 1.57	5.07 ± 1.70	-
Overall mean Δ T for drill design		-	4.37 ± 0.52	3.24 ± 0.48

Δ T: thermal variation.

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
