# Peer review of "In Vitro Study on Bone Heating during Drilling of the Implant Site: Material, Design and Wear of the Surgical Drill"

_materials, 2020, doi:10.3390/ma13081921_

Round 1

Reviewer 1 Report

The work entitled “In vitro study about bone heating during drilling of implant site: Material, design and wear of the surgical drill” appears well written and structured. Anyway, the work does not seem so innovative due to the recent advanced literature on the subject.  Some changes are required to improve the quality of the work.

The authors should introduce some information regarding the wear of the surgical drills, as announced in the title, thus highlighting even if the quality of the eventual drill coating could improve specific features.

As already reported in different works, - C. Ercoli et al. 2004. The influence of drill wear on cutting efficiency and heat production during osteotomy preparation for dental implants: a study of drill durability.  Int. J. Oral Maxillofac. Implants; G. E. Chacon, et al. 2006. Heat production by 3 implant drill systems after repeated drilling and sterilization", J. Oral Maxillofac. Surg; De Santis et al. 2015. An analysis on the inserts for piezoelectric bone surgery: the effect of cutting and sterilization processes. International Journal of Engineering and Innovative Technology (IJEIT) - the cutting efficiency and quality of conventional rotating reamers and drills for bone surgery depend on the wear of rotating tools. Thus, hard coatings are generally implemented on the blades of these rotating tools, especially when precise cuttings and insert stability are required.

Author Response

Dear Reviewer 1:

We appreciate the effort done by the reviewers. Their suggestions will definitely improve the quality of the manuscript.

Reviewer 1:

Reviewer: The authors should introduce some information regarding the wear of the surgical drills, as announced in the title, thus highlighting even if the quality of the eventual drill coating could improve specific features.

Authors: The following statement has been added in the Introduction with the corresponding bibliographic references: Wear due to repeated use of the surgical drill has been recorded thermally in a number of studies [12,16], because the cutting efficiency and quality of conventional rotating drills for bone implant surgery would be affected. Thus, hard coatings are generally applied on the blades of these surgical drills, especially when precise cuttings and insert stability are required. An example is the diamond-like carbon-coated drill. This coating would improve cutting efficiency due to its greater hardness compared to stainless steel [10,11].

A paragraph of the wear is debated in the Discussion:  In the present study, drill wear significantly increased the mean thermal changes after 50 drillings and 10 sterilization cycles in all groups. The thermal change increments in this study were significantly different, in concordance with the literature [30,31]. A study [31] has analyzed wear based on scanning electron microscope analysis, revealing that repeated drill use deteriorates the physical and mechanical surface properties of the surgical drill. This mean thermal increase occurred in all types of materials [9,10,26,32]. The 10 sterilization cycles could be influenced by thermal variations, as evidenced by another study with the use of electron microscopy [33].

Reviewer 2 Report

Here the authors studied the thermal changes during bone drilling by varying the material type of drill (stainless steel vs diamond-like carbon coated), straight vs twisted drill blade, and following different wear. Although I believe the polyurethane medium used should still be complemented with real exvivo bone, I like the authors choice of thermography and robot-mediated drilling to avoid inconsistency. Overall, I would recommend acceptance of the article, following minor clarifications/revisions below:

  • The authors mentioned the overall process lasted 30minutes; how long was each individual drilling? From the different depth for the each drill type, the process lasted for different period of time?
  • How many times were the study repeated for each drill materials and drill types? This would impact the significance value.
  • In Fig 2 & 3, can the authors specify what each dot represents? Also, how were the linear straight lines derived from the dots?

Author Response

Dear Reviewer 2:

We appreciate the effort done by the reviewers. Their suggestions will definitely improve the quality of the manuscript.

Reviewer 2:

Reviewer:The authors mentioned the overall process lasted 30minutes; how long was each individual drilling? From the different depth for the each drill type, the process lasted for different period of time?

Authors: The time of 30 minutes was referred to the time of sterilization, not to the time of the drilling procedure.This has been expressed better: “This sterilization process lasted a total of 30 minutes…”.  Operational procedure was expressed in Table 2: 50mm/min for all the drills and all the depths.

Reviewer:How many times were the study repeated for each drill materials and drill types? This would impact the significance value.

Authors: The following statement has been added in the Material and Method: Each drill was used to perform 50 drillings with a total sample (n) of 600 drillings. The overall mean thermal variations between the SS-SB and DLCC-SB groups was measured as the mean of 200 drillings. The mean thermal variations between the DLCC-SB and DLCC-TB groups was measured as the mean of 100 drillings, since the initial and pilot drills were of the same design and material.

Reviewer: In Fig 2 & 3, can the authors specify what each dot represents? Also, how were the linear straight lines derived from the dots?

Authors: The following legends have been added for each figure:

Figure 3. Thermal evolution over 50 drillings for the drills of the SS-SB and DLCC-SB groups. The green dots represent the thermal variation of each drill of the SS-SB group, and the blue dots represent the thermal variation of each drill of the DLCC-SB group. The green and blue lines correspond to the SS-SB group and DLCC-SB group, respectively, and represent multiple linear regression models of the thermal evolution of these groups.

Figure 4. Thermal evolution over 50 drillings for the drills of the DLCC-SB and DLCC-TB groups. The blue dots represent the thermal variation of the progressive and final drills of the DLCC-SB group, and the red dots represent the thermal variation of the progressive and final drills of the DLCC-TB group. The blue and red lines correspond to the DLCC-SB group and DLCC-TB group, respectively, and represent multiple linear regression models of the thermal evolution of the progressive and final drills of these groups.

Reviewer 3 Report

The topic is interesting, but the proposed research considered the irrealistic possibility of a implant placement without irrigation to prevent the bone heat generation. This implant site preparation  can be performed only in clinical cases with reduced bone density and with a low speed drilling.

Now, the proposed research report a exerimental rotational speed of the surgical drills of 800rpm (no exactly a low speed drilling).

The research was performed using a syntetic polyurethane blocks, which as reported in the Discussion, have a uniform density in opposition to the  clinical real conditions. All residual alveolar crest have cortical and cancellous bone with different density.

Some reported results can be considered obvious:

1) initial and pilot surgical drills generate superior thermal change respect the others because their use is correlate to a maior pressure for perforing the cortical bone

2)It is not described the wear of the different surgical drills after their use. Their thermal variation is uncorrectly correlated only to the design and no to their different materials

3) The reported table not evidence the statistical significance

The discussion is really confused

The correlation between the heat generation and the different surgical drills conductivity is really not clear....

The carbon coating of the diamond-like bone drills is able to assure a reduced wear respect the stailness surgical drills, not a "superior cutting capacity". If it is evidenced, the Authors should report the references

Author Response

(The authors gave the same response as above.)

Reviewer 4 Report

This paper investigates the thermal changes in stainless steel and diamond-like carbon-coated stainless steel drills. They test initial, pilot, progressive, and final drills of both materials, as well as twisted drills and straight for the carbon-coated drills. Thermal changes were measured with thermographic camera during successive drillings of a polyurethane block with uniform density and thickness. The drill material did not show any significant differences in thermal changes between the groups, but the pilot SS drills showed increased thermal changes compared to the carbon-coated pilot drills, and the progressive and final drills had less thermal change than the initial and pilot drills. Twisted carbon-coated drills showed decreased thermal change compared to straight carbon-coated drills. Together, these results indicate that different drill designs can impact thermal changes while drilling without irrigation.   This paper needs major editing for proper English and paragraph form, but it did a good job a setting up the clinical problem and need for assessing thermal changes of drilling without irrigation. Irrigation can block visualization of the osteotomy site and impede harvesting of autologous bone. They completed a well controlled study in vitro. However, it is not clear whether this in vitro setup is relevant to implant dental osteotomy in vivo. As they mentioned, previous studies used bovine bone, with cortical bone and cancellous bone. Here, polyurethane is used of uniform density, and this may also cause an over-estimation of thermal changes, since cortical bone is only 2-3 mm thick, and the underlying cancellous bone is much less dense. Also, the article did not cite relevant thermal properties of polyurethane compared to cortical and cancellous bone, which also may impact thermal changes. They say the mechanical properties are similar, but again, they prepped 11.5 mm into the polyurethane blocks, whereas cortical bone is only 2-3 mm deep. Specific comments are outlined below.
  1. In the methods, please justify why polyurethane blocks are used. How do their thermal properties compare to native bone? Which mechanical properties are similar? Bulk mechanics, like stiffness, is likely less relevant than coefficient of friction, density, and thermal transport properties.
  2. They report thermal changes from camera, but how do they account for thermal changes of the poly-urethane block itself?
  3. The statistical analysis appears sufficient, but sample size and p-values for the specific tests and comparisons between groups should be shown. A column graph with appropriately labelled statistics could show this more clearly than just a table of values. Otherwise, a more complete table should be provided.
  4. Why was a monolithic block of poly-urethane used, rather than two layers of blocks of different densities to mimic layers of cortical and cancellous bone?
  5. Lines 187-190 are confusing. If there are differences in thermal conductivities, then how does this explain that there are no statistically significant differences between materials?
  6. Lines 210-211 are speculative. If the sharpness of carbon-coated blades had a significant effect, then why are there only differences in the pilot blade? I would expect if this is a major factor, then you should see that all the carbon-coated straight blades have less thermal changes than the all the SS straight blades.

Author Response

Dear Reviewer 3:

We appreciate the effort done by the reviewers. Their suggestions will definitely improve the quality of the manuscript.

Reviewer:In the methods, please justify why polyurethane blocks are used. How do their thermal properties compare to native bone? Which mechanical properties are similar? Bulk mechanics, like stiffness, is likely less relevant than coefficient of friction, density, and thermal transport properties.

Authors:

Block was used to reduce variability of the results. Mechanical properties were similar to standard D2 bone according to the Leckhom and Zarb classification. Even though native bone would be more representative of actual reality of the thermal conditions of the osteotomy, main target of the study was to spot differences in the behaviour between experimental groups. Thus, the blocks where considered more appropriate as mechanical and thermal properties are uniform in all of the tests. The following paragraph was added in the Material and Method: The drillings were carried out on standardized synthetic block (Implant training plate, Biomechanicals Models ®, SelModels SL, Barcelona, Spain) with dimensions of 140x90x15 mm. The block was formed by a constant cortical plate of 2 mm of thickness which was covered the cancellous part to simulate human bone conditions type 2 according to the Lekholm and Zarb classification [18]. This structure guaranteed the equivalence of the different drilling sites respect to the density and the ratio between the cortical and cancellous portions [19]. Regarding the thermal conductivity of the standardized synthetic block (0.3 W/m/K) was analogous to the human bone (cortical bone: 0.29 W/m/K and cancellous bone: 0.16-0.34 W/m/K [20]). These thermal conductivity values assured that the measured thermal variation in this block was equivalent to those in human bone [21]. The room temperature was kept constant at 29ºC.

Reviewer:They report thermal changes from camera, but how do they account for thermal changes of the poly-urethane block itself?

Authors:

It was the actual bone block temperature which was measured using the IR camera. IR camera was calibrated (the manufacturer gives a certificate of the accuracy of the calibration) and the temperature displayed was the actual temperature of the block. Other temperature measurement methods have been previously used such as thermocouple wire transductor or thermo resistance, but, the measurement where always affected somehow by thermal inertia of the material surrounding the osteotomy, so the instantaneous temperature was impossible to record with them.

Reviewer:The statistical analysis appears sufficient, but sample size and p-values for the specific tests and comparisons between groups should be shown. A column graph with appropriately labelled statistics could show this more clearly than just a table of values. Otherwise, a more complete table should be provided.

Authors: We agree with you that statistic data could be showed more clearly. For this reason, we have added Figure 2. The legends expose:  Figure 2. Graphic representation of the statistical analysis. The mean thermal variation of each drill and the overall mean thermal variation for each drill material and design are shown.

Reviewer: Why was a monolithic block of poly-urethane used, rather than two layers of blocks of different densities to mimic layers of cortical and cancellous bone?

Authors: We used a block with two layers, but we did not specified in the last manuscript. We added this paragraph in the Material and Method: “The drillings were carried out on standardized synthetic block (Implant training plate, Biomechanicals Models ®, SelModels SL, Barcelona, Spain) with dimensions of 140x90x15 mm. The block was formed by a constant cortical plate of 2 mm of thickness which was covered the cancellous part to simulate human bone conditions type 2 according to the Lekholm and Zarb classification [18]. This structure guaranteed the equivalence of the different drilling sites respect to the density and the ratio between the cortical and cancellous portions [19]. Regarding the thermal conductivity of the standardized synthetic block (0.3 W/m/K) was analogous to the human bone (cortical bone: 0.29 W/m/K and cancellous bone: 0.16-0.34 W/m/K [20]). These thermal conductivity values assured that the measured thermal variation in this block was equivalent to those in human bone [21]. The room temperature was kept constant at 29ºC. And we added another paragraph in the discussion: Regarding the drilling medium, most previously published studies used several drilling media, such as bovine ribs with dissimilar densities and relationships between the cortical and cancellous bone. This fact could affect the resulting thermal variations [21]. Some studies have tried to eliminate this limitation through the use of synthetic blocks [14,23].

Reviewer:Lines 187-190 are confusing. If there are differences in thermal conductivities, then how does this explain that there are no statistically significant differences between materials?

Authors: We agree with you that lines 187-190 were confusing so we have been changed it by this statement: The thermal conductive discrepancy between the tested materials could cause different concentration of the heat at level of friction zone [21] and distinct dissipation of the heat through the drill [10,21,26]. However, these factors appear to be insufficiently influential to show a statistically significant difference in the overall mean thermal variation between the tested materials.

Reviewer:Lines 210-211 are speculative. If the sharpness of carbon-coated blades had a significant effect, then why are there only differences in the pilot blade? I would expect if this is a major factor, then you should see that all the carbon-coated straight blades have less thermal changes than the all the SS straight blades.

Authors: Differences between the tested materials in terms of hardness [10], cutting efficiency [16] or thermal conductivity [25,29] seem to be more important in the most demanding heat-related drilling phases such as the pilot drill (with a combination of lateral drilling and up to maximum drilling depth). These differences may not have a significant impact in less demanding situations such as when the initial, progressive and final drills are used. More studies are needed to elucidate the cause of the lesser thermal effect of the pilot drill, which is the greatest generator of heat.

Round 2

Reviewer 1 Report

Even if the work remains not so innovative due to the recent advanced literature on the subject, it can be accepted in the present form. 

Reviewer 3 Report

The revisioned version of the manuscript give more details about the research design and better descrive its aim. The research is not original, but correct in its procedure.

Reviewer 4 Report

Thank you for responding to the reviewers' comments.

I suggest extensive editing for proper English style, grammar, and paragraph form prior to publication.